# Influence of Sociodemographic, Premorbid, and Injury-Related Factors on Post-Traumatic Stress, Anxiety, and Depression after Traumatic Brain Injury

**DOI:** 10.3390/jcm12123873

**Published:** 2023-06-06

**Authors:** Fabian Bockhop, Katrin Cunitz, Marina Zeldovich, Anna Buchheim, Tim Beissbarth, York Hagmayer, Nicole von Steinbuechel

**Affiliations:** 1Institute of Medical Psychology and Medical Sociology, University Medical Center Göttingen, 37073 Göttingen, Germany; fabian.bockhop@med.uni-goettingen.de (F.B.); katrin.cunitz@med.uni-goettingen.de (K.C.); marina.zeldovich@med.uni-goettingen.de (M.Z.); 2Institute of Psychology, Faculty of Psychology and Sport Science, University of Innsbruck, 6020 Innsbruck, Austria; anna.buchheim@uibk.ac.at; 3Department of Medical Bioinformatics, University Medical Center Göttingen, 37073 Göttingen, Germany; tim.beissbarth@bioinf.med.uni-goettingen.de; 4Georg-Elias-Müller Institute for Psychology, Georg-August-University, 37073 Göttingen, Germany; york.hagmayer@bio.uni-goettingen.de

**Keywords:** post-traumatic stress disorder, generalized anxiety disorder, major depressive disorder, traumatic brain injury, negative and zero-inflated negative binomial model

## Abstract

Psychopathological symptoms are common sequelae after traumatic brain injury (TBI), leading to increased personal and societal burden. Previous studies on factors influencing Post-traumatic Stress Disorder (PTSD), Generalized Anxiety Disorder (GAD), and Major Depressive Disorder (MDD) after TBI have produced inconclusive results, partly due to methodological limitations. The current study investigated the influence of commonly proposed factors on the clinical impairment, occurrence, frequency, and intensity of symptoms of PTSD, GAD, and MDD after TBI. The study sample comprised 2069 individuals (65% males). Associations between psychopathological outcomes and sociodemographic, premorbid, and injury-related factors were analyzed using logistic regression, standard, and zero-inflated negative binomial models. Overall, individuals experienced moderate levels of PTSD, GAD, and MDD. Outcomes correlated with early psychiatric assessments across domains. The clinical impairment, occurrence, frequency, and intensity of all outcomes were associated with the educational level, premorbid psychiatric history, injury cause, and functional recovery. Distinct associations were found for injury severity, LOC, and clinical care pathways with PTSD; age and LOC:sex with GAD; and living situation with MDD, respectively. The use of suitable statistical models supported the identification of factors associated with the multifactorial etiology of psychopathology after TBI. Future research may apply these models to reduce personal and societal burden.

## 1. Introduction

Traumatic brain injury (TBI) is described as an alteration in brain functions as a result of the impact of an external force [1] and has been recognized as a central public health problem deserving of the attention of the world health community [2]. The global incidence of TBI is estimated at 69 million cases per year [3], and a history of TBI is common among general population samples [4]. Thus, the treatment of TBI poses a central challenge for health systems worldwide. While estimates of the incidence and causes of TBI vary across studies and countries, the overall incidence rate of TBI in Europe has been estimated at 262 per 100,000 per year [5], with most cases resulting from incidental falls and road traffic accidents [6]. Other common TBI causes include sports-related injuries [7], assaults [8], or suicidality [9]. The majority of TBI cases are classified as mild (70–90%) [6], most of which do not receive targeted interventions against evolving and/or persistent disabilities [10].

TBI is associated with an increased risk of persistent neurological and psychiatric sequelae (Odds Ratio, OR = 2.00) [7]. The acute care of TBI in itself poses a substantial economic burden [11], but the presence of comorbid psychiatric illness results in a further increase in the cost of treatment, often doubling the total expense [12]. In Europe, individuals after TBI rarely receive psychiatric service during the rehabilitation process despite the apparent need for treatment, e.g., [13]. While the first symptoms of chronic mental illness typically manifest in the post-acute period within the first few weeks after TBI [8], neuropsychiatric impairments occasionally present with a delayed onset from three months to five years post-TBI [9]. Repeated psychiatric and psychopathological assessments starting in the acute injury stage are therefore essential in order to detect signs of emotional disturbance following TBI as soon as possible. Where indicated, patients could then be offered specific psychological and/or medical/pharmaceutical therapeutic interventions.

Mental health concerns are increasingly being addressed by using patient-reported outcome measures (PROMs) as key instruments for the consistent quantification of psychopathological impairment throughout all stages of treatment after TBI. PROM assessments are based on the subjective experience of the individual, independent of expert ratings or technical equipment [14]. Self-report questionnaires have been used to detect clinically significant impairment after TBI with regard to the symptoms of post-traumatic stress disorder (PTSD) [15], general anxiety disorder (GAD) [16], and major depressive disorder (MDD) [17]. The overall incidence rates of PTSD, GAD, and MDD after TBI have been reported to range from 16.5% to 24.5% [18]. Clinical research and treatment may benefit from the regular application of corresponding PROMs to identify clinically significant symptomatology.

The relationship between psychopathological symptoms and TBI is likely based on a multifactorial etiology. Appendix A in the Appendix A (S2 Literary Review Additional Methods and Results) provides an overview of the literature review of factors associated with psychopathological outcomes in adult civilians after TBI [19]. In sum, the previously reported results on the effects of factors potentially associated with psychopathology after TBI remain insufficient and/or inconsistent across different studies and methodologies. This leads to a lack of robust evidence that could be used to inform clinicians and researchers about different types of outcomes and impacting factors. If one’s overall focus lies on the factors contributing to a psychopathological development, the diagnosis or the probability of occurrence of psychopathology may be of primary interest. On the other hand, if one is interested in factors associated with specific diagnostic criteria that can inform prioritizing treatment and rehabilitation, the intensity of symptoms may be more relevant.

The limited generalizability of the above findings is partly due to sample heterogeneity and methodological limitations. A common but often less suitable methodological approach used in the field of TBI is to conduct linear regression analyses to assess the effect of one or more factors on PROM data, e.g., [20]. However, data obtained from relatively short self-report forms are most often strongly skewed to the right, with many subjects endorsing little to no psychopathology [21]. This form of data distribution violates the basic assumptions for linear regression analyses (i.e., normality of residuals) and can result in biased findings [22]. Alternatively, studies have employed logistic regression models to predict the presence of neuropsychiatric impairment after TBI based on clinical thresholds, e.g., [23]. Transforming questionnaire scores into dichotomous outcomes for logistic regression analyses runs the risk of overly reducing data complexity and losing statistical power [24]. Studies also often fail to evaluate the models involved on the basis of appropriate parameters (e.g., area under the receiver–operating characteristic curve, AUC) [25], further limiting the scope of potential predictive effects in logistic regression analyses. These common methodological limitations may highlight the importance of selecting statistical models that reflect the underlying structure of the given data. This may ensure robust conclusions about the association of factors with psychopathology after TBI.

A recent study [26] employed a more specific analytical approach to PROM data collected in a large-scale multicenter project [27], providing robust evidence on the effect of commonly proposed relevant factors on post-concussion symptoms (PCS) after TBI. This method served as the basis for the present investigation.

The aim of the current study was to investigate relevant factors for PTSD, GAD, and MDD after TBI. First, we inspected whether psychopathological domains were associated with one another and to what degree six-month and three-month PROM scores were correlated (i.e., within-subject effects). Second, we examined whether sociodemographic, premorbid, and injury-related factors were associated with the screening diagnoses, occurrence, frequency, and intensity of PTSD, GAD, and MDD after TBI (i.e., between-subjects effects) by applying regression models appropriate for the given data structure.

## 2. Methods

### 2.1. Participant Data

The current study analyzed data extracted from the Collaborative European NeuroTrauma Effectiveness Research in Traumatic Brain Injury (CENTER-TBI) project, which aimed to improve the characterization and clinical treatment of subjects after TBI as part of the European Union (EU) Framework 7 program (EC grant 602150; clinicaltrials.gov NCT02210221) [27]. The CENTER-TBI core study used a prospective observational cohort design and included data on 4509 individuals sampled from 63 medical and research institutions in 18 countries between December 2014 and December 2017. The following inclusion criteria were applied: clinical diagnosis of TBI, institutional presentation within 24 h after injury, clinical indication for computed tomography (CT) scan, and informed consent. Subjects who suffered from pre-existing neurological disorders (e.g., cerebrovascular accident, ischemic attacks, epilepsy) were excluded from the analyses [28].

The current study focused on adult individuals (age ≥ 16 years) across the full spectrum of TBI severity. Participants who had filled out all PROMs at six months (−1/+2 months) after injury were included. See Figure 1 for details of sample attrition.

### 2.2. Ethical Approval

The CENTER-TBI study was conducted in compliance with all relevant laws of the EU that were directly applicable or had a direct effect, as well as all relevant laws of the countries in which the recruitment centers were located, including but not limited to the relevant laws and regulations on the use of human materials, and all relevant guidelines relating to clinical studies, including but not limited to the ICH Harmonized Tripartite Guideline for Good Clinical Practice (CPMP/ICH/135/95) (“ICH GCP”) and the World Medical Association Declaration of Helsinki (“Ethical Principles for Medical Research Involving Human Subjects”). The study attained ethical clearance at each recruitment center, and informed consent for all participants in the CENTER-TBI core study was documented in electronic case report forms (e-CRF, QuesGen Systems Incorporated, Burlingame, CA, USA). A list of recruitment sites, ethics committees, and details on ethical approval can be found on the project’s official website: www.center-tbi.eu/project/ethical-approval (last accessed on 5 February 2022).

### 2.3. Instruments

The data analyzed in the current study were extracted from the CENTER-TBI core 2.1 dataset using the Neurobot platform of CENTER-TBI: https://center-tbi.incf.org (last accessed on 5 February 2022). All instruments used within the CENTER-TBI study, including the PROMs, were translated into the respective languages for use at the recruitment centers according to a standardized protocol, as well as psychometrically validated. For more details, see [29,30]. In addition, recent studies have provided evidence for the equivalence of PROM scores across several languages [31,32]. The respective results demonstrate the comparability of symptoms scores derived from multiple translations of the PROMs, suggesting that differences in scores reflect ‘true’ variance in symptom severity rather than measurement error or bias.

#### 2.3.1. Sociodemographic, Premorbid, and Injury-Related Factors

Sociodemographic (age, sex, living situation, level of education, employment status) and premorbid factors (self-reported psychiatric history prior to the TBI) were collected at study enrollment.

Injury-related factors included the severity of the TBI at baseline. This was clinician-rated using the Glasgow Coma Scale (GCS) [33], also taking into account the presence of abnormalities in the initial CT scans. TBI severity was classified as *uncomplicated mild* (GCS ≥ 13, without CT abnormalities), *complicated mild* (GCS ≥ 13, alongside CT abnormalities), *moderate* (GCS between 9 and 12), and *severe TBI* (GCS ≤ 8). However, it has been reported that the aforementioned classification of *moderate* cases shares various phenotypical (e.g., intracranial hypertension, cerebral contusions, diffuse axonal injury) and treatment (e.g., ICU admission, serial examinations, serial CT scans, neurosurgical consultations) characteristics with *severe* injuries [34]. Therefore, the current study utilized a combined severity class. The final classification of TBI severity was *uncomplicated mild*, *complicated mild*, and *moderate/severe*.

Functional recovery after TBI was clinician-rated using the extended Glasgow Outcome Scale (GOSE) [35] on an eight-point scale (1: dead, 2: vegetative state, 3–4: severe disability, 5–6: moderate disability, 7–8: good recovery). For more details on the extraction of GOSE data, see [29].

The severity of physical symptoms after TBI was rated using the Injury Severity Score (ISS). Individual ISS can range from 0 to 75, and higher values indicate greater impairment.

Finally, medical records were used to assess information on the injury cause; loss of consciousness (LOC) during the TBI event; and the clinical care pathway where the TBI and/or extracranial injuries were treated (i.e., emergency room, ward, ICU).

#### 2.3.2. PROM Screenings for Psychopathological Symptoms

The Post-traumatic Stress Disorder Checklist-5 (PCL-5) [36] is a 20-item PROM screening for symptoms of PTSD. Individuals indicate impairment caused by each symptom during the past month on a five-point Likert scale ranging from 0 (not bothered at all) to 4 (extremely bothered). Total scores are calculated ranging from 0 to 80, and a clinical screening cutoff may be applied at 31 [37].

The Generalized Anxiety Disorder 7-Item Scale (GAD-7) [38] is a seven-item PROM screening for symptoms of GAD during the past two weeks. It utilizes a four-point Likert scale ranging from 0 (*not bothered at all*) to 3 (*bothered nearly every day*). The total score ranges from 0 to 21, with the cutoff for clinical screening at 10 or above [38].

The Patient Health Questionnaire 9 (PHQ-9) [39] is a nine-item PROM screening for symptoms of MDD. The presence of each symptom over the course of the past two weeks is rated on a four-point Likert scale from 0 (*not bothered at all*) to 3 (*bothered nearly every day*). The total score ranges from 0 to 27, and the clinical cutoff is set at 10 or above [39].

### 2.4. Statistical Analyses

#### 2.4.1. Correlational Analyses

The relationship between the PROM scores across psychopathological domains was investigated using Spearman rank correlations for dependent data on the six-month PCL-5, GAD-7, and PHQ-9 total scores. Similarly, in order to examine the relationship between the three-months and six-months psychopathological screenings, Spearman rank correlations for dependent data were carried out for individuals who completed both assessments. The results of the correlational analyses are presented in Appendix A in the Appendix A (S2 Literary Review Additional Methods and Results).

#### 2.4.2. Regression Models

The association of the above factors with the different psychopathological outcomes after TBI was investigated using three types of regression models. Table 1 provides an overview. The methodological approaches are described in detail in the Appendix A (S2 Literary Review Additional Methods and Results). First, logistic regressions (LR) were used to examine which factors were associated with the development of psychopathology after TBI (e.g., positive screening for GAD). Second, zero-inflated negative binomial models (ZINB), a relatively new approach, investigated the number of symptoms developed with regard to a psychopathological domain (e.g., no symptoms of GAD vs. three out of seven symptoms of GAD rated as being at least mild). This method supports a diagnostic perspective, since the presence of symptoms is the basis for the clinical diagnosis. Finally, standard negative binomial models (NB) account for non-normal data distributions when examining PROM scores. The results show which factors are associated with more intense psychopathology (e.g., higher GAD-7 scores).

All analyses were conducted in R 4.1.0 [40] using the packages *dplyr* [41], *pscl* [42], and *MASS* [43]. The level of significance in analyses was set at 5%.

## 3. Results

### 3.1. Sample Characteristics

The final sample consisted of *N* = 2069 individuals who completed all PROMs six months after experiencing a TBI. The participants were predominantly male (65.00%) and had a mean age of 49.09 years (*SD* = 19.32). While only a minority of participants sustained a moderate/severe TBI (22.81%), individuals were frequently admitted to the ICU (40.60%). This was likely due to the pronounced extracranial injuries (Total ISS, *M* = 18.58, *SD* = 15.00), since individuals with high extracranial injuries (Total ISS ≥ 15) most commonly received treatment in the ICU (77.81%). Most participants showed good recovery (64.57%). Psychopathological screenings revealed subgroups with clinically relevant PTSD (10.83%), GAD (10.78%), and MDD (17.74%). Table 2 provides an overview of the descriptive statistics, including the sociodemographic, premorbid, and injury- and outcome-related characteristics.

Importantly, a sizeable number of individuals were not eligible for inclusion in the current study due to a lack of complete psychopathological screenings at six months post-TBI (*N* = 1040). Comparisons of the descriptive characteristics of the subjects included in the final sample with those of excluded individuals revealed that the latter group tended to be younger (*M* = 48.09, *SD* = 20.53), was more likely to be admitted to the ICU (50.00%), reported more severe extracranial injuries in the Total ISS (*M* = 20.5, *SD* = 15.41), and was less likely to have experienced a good recovery (55.67%). In sum, evidence suggests that individuals who did not complete psychopathological screenings at six months after TBI experienced more severe disability. However, renewed analyses for individuals who completed at least one PROM at six months post-TBI (PCL-5: *N* = 2116; GAD-7: *N* = 2122; PHQ-9: *N* = 2125) produced the same overall results in the main analyses, with only minor deviations (see Appendix A in the Appendix A Literary Review Additional Methods and Results).

### 3.2. Regression Models

The LR models targeted the effect of factors on the psychopathological screening diagnoses after TBI. Table A1, Table A2 and Table A3 in Appendix B present the coefficients of the sociodemographic, premorbid, and injury-related factors associated with clinical levels of PTSD, GAD, and MDD in LR models. Figure 2 provides a visualization of the effects of the factors on the psychopathological screening diagnoses, depicting Odds Ratios (ORs) with 95%-CIs. The diagnosis of PTSD was associated with no school or primary school education, a premorbid history of psychiatric symptoms, having experienced a road traffic accident or other injury cause besides incidental falls, a loss of consciousness, as well as moderate and severe disability. In contrast, PTSD diagnoses were less common in individuals who had experienced a moderate or severe TBI. Preliminary GAD diagnoses were associated with female sex, none or primary, secondary, or high school education, premorbid psychiatric symptoms, TBI caused by a road traffic accident, as well as moderate and severe disability following TBI. GAD diagnoses were less common in individuals with a high number of extracranial injuries. MDD screening diagnoses were associated with female sex, no or primary school education, unemployment, premorbid psychiatric symptoms, as well as moderate and severe disability after TBI. In contrast, the significant interaction between LOC and sex indicated that LOC was associated with a lower probability of a preliminary MDD diagnosis in females compared to males. All significant effects were between 8.84 ≥ OR ≥ 1.65 and 0.40 ≤ OR ≤ 0.98. No significant effects (*p* > 0.05) were observed in any of the LR models for age, living status, clinical care pathway, or the interactions between sex and age. Nagelkerke’s R^2^ for the LR models was 0.21 ≥ R^2^ ≥ 0.16, and the AUC values were 0.77 ≥ AUC ≥ 0.74, indicating a high model fit and the correct classification of 74% to 77% of individuals with regard to psychopathological screening diagnoses.

Table A4, Table A5 and Table A6 in Appendix C present the coefficients of the sociodemographic, premorbid, and injury-related factors associated with symptoms of PTSD, GAD, and MDD in the ZINB models. Figure 3 depicts the effects of the factors on the psychopathological symptoms, using ORs for the zero part in the left panel and Rate Ratios (RRs) for the count part in the right panel of the respective ZINB models. Table A7, Table A8 and Table A9 in Appendix D depict the coefficients of the sociodemographic, premorbid, and injury-related factors associated with PROM scores in NB models. Figure 4 shows the effects of the factors on the PROM scores. Finally, Table 3 provides an overview of the findings on the sociodemographic, premorbid, and injury-related factors associated with the diagnoses as well as the occurrence, frequency, and intensity of symptoms of PTSD, GAD, and MDD across all models.

The zero parts of the ZINB models (Figure 3, left panel) investigated the effect of the factors on the probability of the absence compared with the presence of psychopathological symptoms. The ZINB analyses revealed that an absence of PTSD symptoms was less likely in individuals who experienced road traffic accidents and individuals who were admitted to a hospital ward. The ZINB model for GAD symptoms showed that an absence of GAD symptoms was less likely for no or primary school education, unemployment before TBI, a history of premorbid psychiatric symptoms, and the experience of moderate or severe disability following TBI. By contrast, the absence of GAD symptoms was positively associated with older age (OR = 1.02; CI_95%_ (1.00, 1.03); *p* = 0.01). Finally, the absence of MDD symptoms was less likely in individuals with premorbid psychiatric symptoms and with moderate disability after TBI. All significant effects were between 0.18 ≤ OR ≤ 0.58. No significant effects (*p* > 0.05) were observed in the zero part of any of the ZINB models for sex, living situation, injury severity, extracranial injuries, LOC, as well as the interactions between sex and age or sex and LOC.

The count parts of the ZINB models (Figure 3, right panel) targeted the effect of factors on the number of psychopathological symptoms. Individuals experienced more PTSD symptoms if they had no or only primary school education, had a history of premorbid psychiatric symptoms, sustained the TBI as a result of a road traffic accident or other injury cause, as well as had moderate or severe disability. By contrast, individuals after moderate or severe TBI experienced a lower number of PTSD symptoms. The frequency of GAD symptoms was significantly associated with female sex, no or primary school education, a history of psychiatric symptoms, having sustained a TBI after a road traffic accident or other injury cause, and the indication of moderate or severe disability. Here, the significant interaction between LOC and sex showed that the experience of LOC was associated with fewer GAD symptoms in females compared to males. Finally, more pronounced MDD symptoms were associated with no or primary school education, living alone, a history of premorbid psychiatric symptoms, having experienced a TBI as a result of a road traffic accident, as well as moderate or severe disability. All significant effects were between 1.84 ≥ RR ≥ 1.10 and 0.68 ≤ RR ≤ 0.83. No significant effects (*p* > 0.05) were observed for age, LOC, extracranial injuries, the clinical care pathway, and the age–sex interaction.

The NB models targeted the effect of factors on the intensity of PROM scores following TBI. Figure 4 shows the effects of the factors on the psychopathological PROM scores using RRs with 95%-CIs. Higher PCL-5 scores were associated with female sex, no or primary school education, a premorbid history of psychiatric symptoms, having experienced a road traffic accident or other injury cause, loss of consciousness, as well as moderate and severe disability. High PCL-5 scores were less common in individuals who had experienced a moderate or severe TBI (RR = 0.75; CI_95%_ (0.60, 0.95); *p* = 0.01). Higher GAD-7 scores were associated with female sex, no or primary school education, premorbid psychiatric symptoms, TBI caused by a road traffic accident, as well as moderate and severe disability following TBI. Higher PHQ-9 scores were associated with female sex, none or primary school education, unemployment, living alone, premorbid psychiatric symptoms, as well as moderate and severe disability after TBI. All significant effects were between 2.75 ≥ RR ≥ 1.21. No significant effects (*p* > 0.05) were observed in any of the NB models for age, extracranial injuries, the clinical care pathway, and the interactions between sex and age or sex and LOC.

## 4. Discussion

The current study aimed to examine factors associated with symptoms of post-traumatic stress disorder, generalized anxiety disorder, and major depressive disorder following TBI. Previous prediction studies have produced inconclusive results on the effect of specific factors causing psychopathological impairment in individuals after TBI, in part due to methodological limitations, e.g., [23]. The current study was the first to use an integrated analytic approach particularly suited for the given data structure. While certain factors were associated only with the clinical impairment, the occurrence, the frequency, or the intensity of symptoms of PTSD, GAD, or MDD, other factors were notably related to multiple outcomes. Robust evidence on factors significantly associated with the psychopathology after TBI is a highly valuable source of information for targeted clinical interventions and can help to reduce both the personal burden and the societal healthcare costs worldwide.

### 4.1. Within-Subject Factors

The study sample as a whole showed moderate levels of PTSD, GAD, and MDD at six months after TBI (10.78–17.74%), which were somewhat lower than previously reported incidence rates (16.5–24.5%) [18]. Analyses of within-subject effects (see Appendix A in Appendix A Literary Review Additional Methods and Results) revealed strong associations between symptoms of PTSD, GAD, and MDD assessed at three and six months post TBI. Our results show that symptoms in one psychopathological domain can serve as markers for impairment in other domains due to the comorbidity of PTSD, GAD, and MDD. Moreover, our findings indicate that psychopathological screenings at relatively early timepoints after TBI can be useful in predicting later psychopathological impairment. Although the current study sample overall displayed comparatively low levels of psychopathological impairment at three as well as six months after TBI, with no signs of delayed-onset psychopathology [9], the manifestation of later clinical impairment is most often preceded by sub-threshold symptoms [44]. Consequently, the application of multiple validated psychopathological screening questionnaires at early up to later timepoints after TBI should be considered for inclusion into guidelines for the management of TBI, e.g., [45].

### 4.2. Between-Subjects Factors

The current study examined the effects of several sociodemographic, premorbid, and injury-related between-subject factors on PTSD, GAD, and MDD following TBI. Among the sociodemographic factors, a low educational status was significantly and consistently associated with more pronounced psychopathological outcomes after TBI. Moreover, female sex was associated with psychopathological outcomes, particularly with regard to GAD and MDD. These results replicate some of the previous findings which commonly report sex and education to be associated with PTSD [46], GAD [47], and MDD [48] after TBI (see Appendix A in Appendix A Literary Review Additional Methods and Results). Furthermore, in the current study, unemployment was related particularly to MDD and GAD. This is consistent with previous evidence describing an association of unemployment with anxiety and depression after TBI [49]. By contrast, previous research showed no significant effect of employment on PTSD after TBI [50]. Living alone was mainly associated with MDD in the present study sample, underlining prior results, e.g., [51]. Interestingly, little to no evidence was found for a relation of age or an age:sex interaction with any of the psychopathological outcomes. This underlines the findings of other studies, e.g., [50]. Age was associated only with the occurrence of GAD symptoms, which may reflect an association of older age with anxiety after TBI, e.g., [52]. Overall, our results suggest that, particularly, the sex and education of individuals should be considered as markers for a generally increased risk of psychopathology after TBI. Other sociodemographic factors (e.g., age) may play a more specific role for distinct psychopathological domains.

Importantly, in the current study, individuals’ premorbid psychiatric history was among the factors with the strongest associations with all psychopathological outcomes after TBI. Previous research has already found that premorbid mental health problems are among the most important risk factors for depression and PTSD [53] as well as persistently high anxiety after TBI [54] (see Appendix A in Appendix A Literary Review Additional Methods and Results). Despite the fact that the subjective recall of one’s self-reported psychiatric history does not always correspond directly with the respective medical records [55], our results suggest that the time- and cost-effective assessment of individuals’ previous psychiatric problems may be a highly valuable source of information for clinical care after TBI.

With regard to injury-related factors, individuals’ functional recovery status and the injury cause were most consistently and strongly correlated with the psychopathological outcomes following TBI. Our results reinforce previous findings showing an association of individuals’ functional status (i.e., GOSE) with PTSD and depression [56] as well as anxiety [57] (see Appendix A in Appendix A Literary Review Additional Methods and Results). The strong association between psychopathological outcomes and functional recovery is unsurprising given that the clinical GOSE interview assesses disability in multiple life domains (e.g., return to normal life, family and friends, social and leisure activities) that may be impacted by psychopathological symptoms. Therefore, the added value of PROM scores lies particularly in providing a short and economic assessment of individuals’ subjective perspective of impairment. Furthermore, the effect of the cause of injury found in the current study underlines the importance of injury mechanisms in the treatment of TBI. However, our results indicate a relatively low psychopathology in a sample that most commonly experienced incidental falls (43.50%). Psychopathology seems particularly prevalent after experiencing traumatic events such as RTAs [58]. Consequently, the injury cause should be routinely assessed, suggesting particularly close clinical monitoring of individuals whose injury resulted from causes other than incidental falls.

The current study found that experiencing a more severe TBI was only associated with lower overall levels of PTSD. However, previous research had produced controversial results concerning the effect of TBI severity on PTSD. The emergence of PTSD and PTSD-like symptoms has been described after mild [59] as well as after severe TBI [60]. Interestingly, our findings underline evidence suggesting that the experience of severe TBI can be protective against the development of PTSD symptoms [61]. This is surprising because only about a fifth of the individuals after TBI in the present study sample were classified as moderate/severe (22.81%). In addition, our results have implications for the categorization of TBI severity. We differentiated cases into complicated mild or uncomplicated mild based on CT scans, as well as moderate/severe. Despite the limited impact of TBI severity on the psychopathological symptoms observed in our study, previous research has demonstrated the utility of this severity classification in predicting outcomes such as mortality [34] after TBI. Thus, the accessibility and affordability of brain imaging technologies for treatment centers as diagnostic tools for potential structural damage after TBI should be facilitated.

Little or no evidence was found for an effect of LOC, an LOC:sex interaction, extracranial injuries, or individuals’ clinical care pathways on any of the psychopathological outcomes following TBI. This reflects the common debate on the association of LOC with PTSD after TBI [62] and is consistent with findings showing that extracranial injuries are not related to PTSD and depression after mild to moderate TBI [63]. Moreover, while the lack of an effect of clinical care pathways (i.e., ER, ward, ICU) found in the present work is contrary to previous results [64], this may explain why relatively few studies consider the clinical care pathway to be a relevant factor for psychopathological outcomes after TBI (see Appendix A in Appendix A Literary Review Additional Methods and Results).

Our main analyses used LR, ZINB, and NB models to investigate whether factors were specifically associated with either screening diagnoses, the presence and severity of psychopathology following TBI, or a combination of these. Thus, our findings underline the importance of distinguishing factors associated with the clinical impairment, the occurrence, the frequency, and the intensity of psychopathological symptoms after TBI. Clinical diagnostics determine the number and composition of symptoms concerning PTSD, GAD, and MDD. The statistical models employed (i.e., the count part in ZINB models) allow for the detection of patient characteristics associated with experiencing a greater number of symptoms and more intense symptoms and, hence, a greater likelihood of a clinical diagnosis.

Overall, the use of ZINB models led to similar results to those of the more conventional LR and NB models with regard to psychopathology after TBI. However, whereas the LR and NB models, for instance, showed that females experienced more intense psychopathological outcomes, no sex differences were found for the occurrence or frequency of psychopathological symptoms following TBI. This somewhat contradictory finding reflects the conflicting results as to whether females were more [65] or less [66] likely to receive a psychiatric diagnosis after TBI. As the present study suggests, the assessment of psychopathology based on clinical cutoffs or PROM scores may have led to the conclusion of higher rates of PTSD, GAD, and MDD in females after TBI compared with clinical symptoms.

In addition, future studies should increasingly apply multivariate (vs. univariate) models for the simultaneous effects of factors on multiple psychopathological outcomes after TBI. Multivariate analyses generally allow complex relationships between independent and dependent variables to be investigated, resulting in more realistic conclusions with regard to their clinical impact compared with univariate techniques. While multivariate linear models have been used before in prediction studies in the field of TBI, e.g., [67], more specific approaches such as multivariate NB or ZINB models are not yet well-established e.g., [68].

### 4.3. Strengths and Limitations

The current study has a number of strengths. First, we were able to utilize data (*N* > 2000) from a large-scale multicenter study in 18 countries including individuals across the entire severity spectrum of TBI who suffered from moderate levels of PTSD, GAD, and MDD. Second, this study is among the first to combine estimates of screening diagnoses as well as the occurrence, frequency, and intensity of psychopathological outcomes concurrently. The differences in the results between the respective models may explain some of the contradictions in the existing literature on the factors associated with PTSD, GAD, and MDD. Consequently, this statistical approach allows more robust conclusions to be drawn about the relationship between factors and psychopathological outcomes after TBI.

Despite the aforementioned strengths, our study had some limitations. First, since the current overall study sample was middle-aged and had experienced a TBI after incidental falls or RTA, the generalizability of the present results to other populations may be limited. Hence, future replication studies should target other TBI as well as non-TBI trauma-affected samples (e.g., pediatric, elderly, military personnel, and survivors of natural disasters and domestic or sexual violence) in order to more broadly evaluate the effect of factors associated with psychopathology.

Second, the LR, NB, and ZINB models applied have not yet been externally validated via statistical methods such as bootstrapping, e.g., [69]. Instead, models were compared to and verified based on the respective companion models following a previously proposed analytic approach [26]. However, they displayed a significantly better fit compared with intercept-only models. Thus, we concluded that the regression models had been sufficiently validated for the purposes of the current study.

Third, some additional sociodemographic and clinical variables were not available for the current analyses and should be considered in future studies (e.g., ethnicity, clinical diagnoses of psychiatric disorders, family history of psychiatric disorders, cumulative traumatic experiences) [46].

Finally, biological factors of psychopathology such as biomarkers, e.g., [70], could largely not be accounted for. However, the effect of these individually is rather small (e.g., 70). Consequently, the collection of blood-serum or imaging samples for assessing single relevant factors associated with psychopathology after TBI may prove excessively costly for some treatment centers.

## 5. Conclusions

Individuals after TBI are at an elevated risk of suffering from psychopathological symptoms, which calls for the use of appropriate assessment methods, diagnostic tools, and treatment interventions. Our use of combined multiple complementary statistical approaches provides nuanced insights into the role of specific factors affecting the development, the prevalence, and the severity of psychopathology following TBI. Research on the mechanisms of various sociodemographic, premorbid, and injury-related factors associated with the presence of psychopathology plays an important role in the early identification of high-risk individuals in need of therapeutic intervention. The scientific implications of the present study include investigating influential factors with regard to several types of assessment (e.g., screening diagnoses, symptoms, PROM scores) of psychopathological outcomes after TBI. In addition, alternative statistical models (e.g., multivariate zero-inflated negative binomial regression) should become technologically and commercially available. On the basis of our results, societal and clinical implications include targeted mental health programs for particularly vulnerable populations after TBI (e.g., females, low educational status), close psychopathological monitoring of TBI patients with a history of psychiatric symptoms, and an enhancement of safety features to prevent road traffic accidents with TBI (e.g., collision avoidance systems, airbag technology). This is likely to improve targeted clinical therapy, care, rehabilitation, and research after TBI, thereby reducing the personal health burden on individuals as well as the cost of treatment.

## Figures and Tables

**Figure 1 jcm-12-03873-f001:**
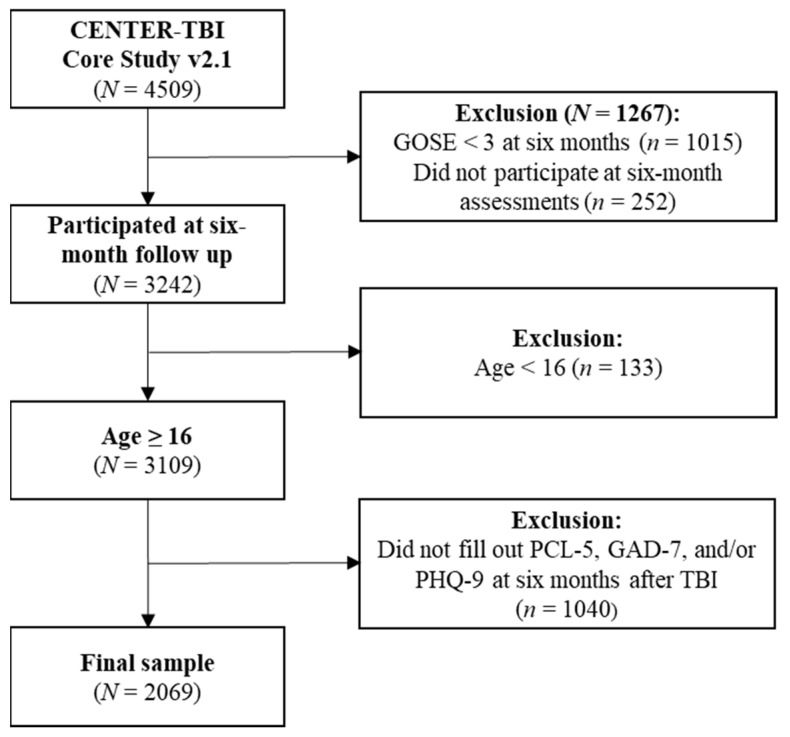
Sample attrition diagram of the total sample included in the current study. TBI = traumatic brain injury, GOSE = Glasgow Outcome Scores Extended, PCL-5 = Post-traumatic Stress Disorder Checklist for DSM-5, GAD-7 = Generalized Anxiety Disorder Scale, PHQ-9 = Patient Health Questionnaire.

**Figure 2 jcm-12-03873-f002:**
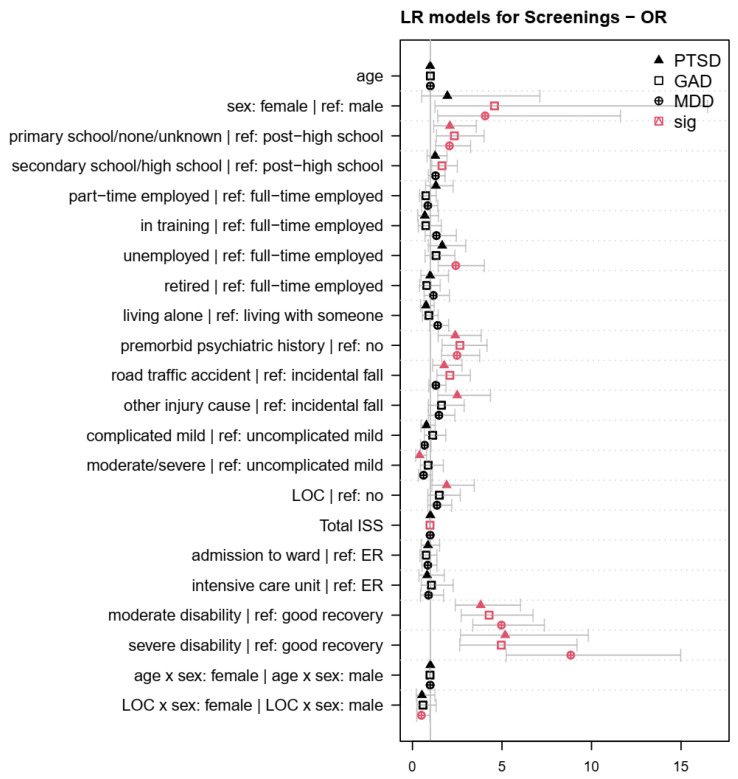
Odds ratios (OR) for sociodemographic, premorbid, and injury-related factors associated with screening diagnoses in the logistic regression (LR) models. Red symbols indicate factors, factor levels, or interactions with a significant effect on the psychopathological screening diagnoses (*p* < 0.05). Values > 1 indicate that factors were associated with an increased probability of a psychopathological screening diagnosis compared with the reference group (for nominal variables) or higher values (for continuous variables).

**Figure 3 jcm-12-03873-f003:**
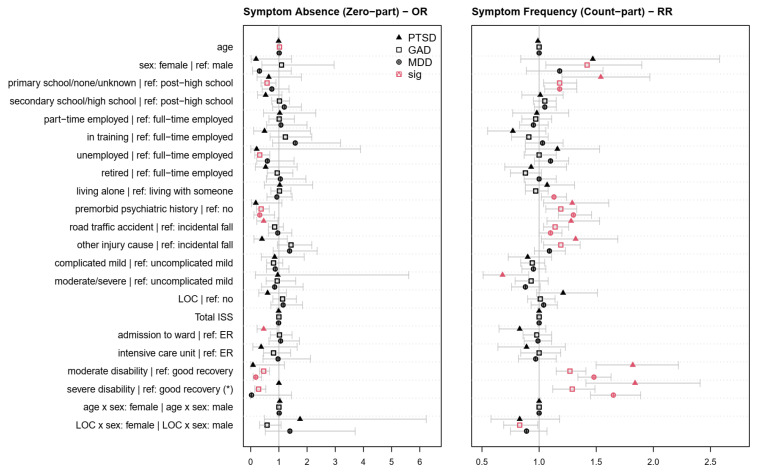
Odds ratios (OR (**left panel**)) and rate ratios (RR (**right panel**)) for sociodemographic, premorbid, and injury-related factors associated with psychopathological symptoms in the final ZINB models. Red symbols indicate factors, factor levels, or interactions with a significant effect on the occurrence (**left panel**) and frequency (**right panel**) of PTSD, GAD, or MDD symptoms (*p* < 0.05). * Some analyses could not be calculated due to insufficient variability. The zero part indicates the probability of the absence of psychopathological symptoms, i.e., values < 1 indicate that the probability of developing symptoms is increased compared with the reference group (for nominal variables) or for higher values (for continuous variables). The count part indicates the probability of developing more psychopathological symptoms, on average, i.e., values > 1 indicate an increased probability of developing more symptoms of PTSD, GAD, or MDD compared with the reference group (for nominal variables) or with higher values (for continuous variables).

**Figure 4 jcm-12-03873-f004:**
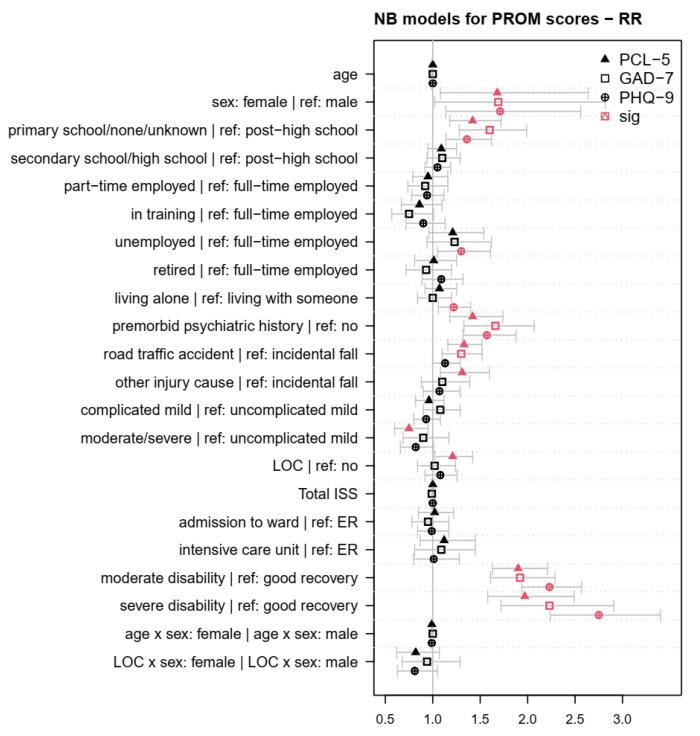
Rate ratios (RR) for sociodemographic, premorbid, and injury-related factors associated with PROM scores in the final negative binomial (NB) models. Red symbols indicate factors, factor levels, or interactions with a significant effect on the intensity of psychopathological symptoms (*p* < 0.05). Values > 1 indicate that factors were associated with higher PROM scores compared with the reference group (for nominal variables) or with higher values (for continuous variables).

**Table 1 jcm-12-03873-t001:** Characteristics of the regression models used in the current study.

Type of Regression	Outcome	Interpretation	Scale of Data	Dependent Variable	Index
Logistic (LR)	Screening Diagnoses	Effect of factors on the probability of a psychopathological screening diagnosis	Nominal (0: absent, 1: present)	Clinical cutoffs	OR (95%-CI)
Zero-inflated Negative Binomial (ZINB)	Occurrence and Frequency of Symptoms	Effect of factors on the probability of the occurrence and frequency of psychopathological symptoms	Occurrence: Nominal Frequency: Count (0–*k*, where *k* is the maximum number of items in a PROM)	Number of symptoms rated as at least mild	Occurrence: OR (95%-CI) Frequency: RR (95%-CI)
Negative Binomial (NB)	Intensity of PROM scores	Effect of factors on the intensity of psychopathological symptoms	Metric (0–*m*, where *m* is the maximum scale score in a PROM)	Raw PROM score	RR (95%-CI)

Note. PROM = Patient-Reported Outcome Measure (i.e., PCL-5, GAD-7, PHQ-9), OR = Odds Ratio, CI = Confidence Interval, RR = Rate Ratio.

**Table 2 jcm-12-03873-t002:** Sociodemographic, premorbid, and injury- and outcome-related characteristics of the current study sample.

	Group (Reference in *Italics*)	*N* (%)	*M* (*SD*)
Age	-	2069 (100)	49.09 (19.32)
Sex	*male*	1352 (65.35)	-
	female	717 (34.65)	-
Education	*college/university*	967 (46.74)	-
	secondary/high school	628 (30.35)	-
	none/primary school	258 (12.47)	-
	missing	216 (10.44)	-
Employment	*full-time employed*	883 (42.68)	-
	part-time employed	225 (10.87)	-
	in training	193 (9.33)	-
	unemployed	163 (7.88)	-
	retired	477 (23.05)	-
	missing	128 (6.19)	-
Living situation	*with someone*	1651 (79.80)	-
	alone	417 (20.15)	-
	missing	1 (0.05)	-
Premorbid psychiatric history	*no*	1803 (87.14)	-
	yes	244 (11.79)	-
	missing	22 (1.06)	-
Injury cause	*incidental fall*	900 (43.50)	-
	road traffic accident	844 (40.79)	-
	other	286 (13.82)	-
	missing	39 (1.88)	-
TBI severity	*uncomplicated mild*	653 (31.56)	-
	complicated mild	618 (29.87)	-
	moderate/severe	472 (22.81)	-
	missing	326 (15.76)	-
Loss of consciousness	*no*	641 (30.98)	-
	yes	1223 (59.11)	-
	missing	205 (9.91)	-
Extracranial injuries (ISS)	-	2048 (98.99)	18.58 (15.00)
Clinical care pathways	*ER*	437 (21.12)	-
	Ward	792 (38.28)	-
	ICU	840 (40.60)	-
Recovery (GOSE)	*good recovery*	1336 (64.57)	-
	moderate disability	198 (9.57)	-
	severe disability	534 (25.81)	-
	missing	1 (0.05)	-
PCL-5	3 M	1630 (78.78)	12.92 (13.57)
	6 M	2069 (100)	12.25 (13.64)
GAD-7	3 M	1618 (78.20)	3.63 (4.46)
	6 M	2069 (100)	3.59 (4.49)
PHQ-9	3 M	1626 (78.59)	5.20 (5.21)
	6 M	2069 (100)	5.00 (5.30)
PTSD Screening Diagnosis	No	1845 (89.17)	-
	Yes	224 (10.83)	-
GAD Screening Diagnosis	No	1846 (89.22)	-
	Yes	223 (10.78)	-
MDD Screening Diagnosis	No	1702 (82.26)	-
	Yes	367 (17.74)	-
Total	-	2069 (100)	-

Note. TBI = traumatic brain injury, ISS = total injury severity score, ER = emergency room, ICU = intensive care unit, GOSE = Glasgow Outcome Scores Extended, PCL-5 = PCL-5 total score, GAD-7 = GAD-7 total score, PHQ-9 = PHQ-9 total score, 3 M = three-month assessment, 6 M = six-month assessment. Screening diagnoses are assessed based on the commonly used cutoffs for PTSD (i.e., PCL-5 ≥ 31), GAD (i.e., GAD-7 ≥ 10), and MDD (i.e., PHQ-9 ≥ 10).

**Table 3 jcm-12-03873-t003:** Overview of the sociodemographic, premorbid, and injury-related factors associated with PTSD, GAD, and MDD.

		PTSD				GAD				MDD			
No.	Factor	Screening ^a^	Intensity ^b^	Occurrence ^c^	Frequency ^d^	Screening ^a^	Intensity ^b^	Occurrence ^c^	Frequency ^d^	Screening ^a^	Intensity ^b^	Occurrence ^c^	Frequency ^d^
(1)	Age							**					
(2)	Sex		*			*	*		*	**	**		
(1:2)	Age:Sex			†									
(3)	Education	**	***	†	***	***	***	*	**	***	***		**
(4)	Employment	†					†	***	†	***	*		
(5)	Living Situation									†	***		*
(6)	Premorbid Psychiatric History	***	***	†	*	***	***	***	***	***	***	*	***
(7)	Injury Cause	***	***	*	**	***	***	†	**		*		*
(8)	Injury Severity	**	**		*					†	†		†
(9)	LOC		*		†								
(9:2)	LOC:Sex							†	*	*	†		
(10)	Extracranial Injuries (ISS)					*	†						
(11)	Clinical Care Pathway (ER, ADM, ICU)			*									
(12)	Recovery	***	***	†	***	***	***	***	***	***	***	***	***

Note. PTSD = Post-traumatic Stress Disorder, GAD = Generalized Anxiety Disorder, MDD = Major Depressive Disorder, LOC = loss of consciousness, ISS = total injury severity score, ER = emergency room, ADM = admission to ward, ICU = intensive care unit; Age:Sex = interaction term, LOC:Sex = interaction term. ^a^ The association of the factors with the psychopathological screening diagnoses is analyzed in the logistic regression (LR) models; ^b^ The association of the factors with the intensity of PROM scores is analyzed in the negative binomial (NB) models; ^c^ The association of the factors with the occurrence of symptoms of PTSD, GAD, and MDD is analyzed in the zero part of the zero-inflated negative binomial (ZINB) models; ^d^ The association of the factors with the frequency of symptoms of PTSD, GAD, and MDD is analyzed in the count part of the ZINB models; asterisks (*) indicate the most pronounced significant association per factor; † *p* < 0.10; * *p* < 0.05; ** *p* < 0.01; *** *p* < 0.001.

## Data Availability

All relevant data are available upon request from CENTER-TBI, and the authors are not legally allowed to share it publicly. The authors confirm that they received no special access privileges to the data. CENTER-TBI is committed to data sharing and in particular to responsible further use of the data. Hereto, we have a data sharing statement in place: https://www.center-tbi.eu/data/sharing (accessed on 5 February 2022). The CENTER-TBI Management Committee, in collaboration with the General Assembly, established the Data Sharing policy and Publication and Authorship Guidelines to assure correct and appropriate use of the data as the dataset is hugely complex and requires help of experts from the Data Curation Team or Bio- Statistical Team for correct use. This means that we encourage researchers to contact the CENTER-TBI team for any research plans and the Data Curation Team for any help in appropriate use of the data, including sharing of scripts. Requests for data access can be submitted online: https://www.center-tbi.eu/data (accessed on 5 February 2022). The complete Manual for data access is also available online: https://www.center-tbi.eu/files/SOP-Manual-DAPR-2402020.pdf (accessed on 5 February 2022).

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
