# Peer review of "Influence of Sociodemographic, Premorbid, and Injury-Related Factors on Post-Traumatic Stress, Anxiety, and Depression after Traumatic Brain Injury"

_jcm, 2023, doi:10.3390/jcm12123873_

Round 1

Reviewer 1 Report

Dear authors,

your paper - Influence of Sociodemographic, Premorbid, and Injury-Related Factors on Post-traumatic Stress, Anxiety, and Depression after Traumatic Brain Injury is a very significant contribution to this field. On the other side, I have some comments:

1. Introduction part must be improved with literary review part which is very important for this study;

2. Sample characteristics move in a methods part with the title - sample...

3. In the results part, I would suggest you move the regression part to the beginning of the paper;

4. Please, improve the conclusion part with more scientific and social implications;

Other parts of the study are very nicely written. 

Kind regards

Minor editing of English language required

Author Response

Answer to Reviewer I concerning the manuscript

Influence of Sociodemographic, Premorbid, and Injury-Related Factors on Post-traumatic Stress, Anxiety, and Depression after Traumatic Brain Injury

Dear authors,

your paper – Influence of Sociodemographic, Premorbid, and Injury-Related Factors on Post-traumatic Stress, Anxiety, and Depression after Traumatic Brain Injury – is a very significant contribution to this field. On the other side, I have some comments:

  1. Introduction part must be improved with literary review part which is very important for this study;

Response: Dear reviewer, thank you very much for feedback. We agree that presenting a literary review is very important for the current study. Due to the large number of relevant studies, we decided to present the literature in the form of a table in the Online Supplement (OS-1 Literary Review). This was done for the sake of brevity (as emphasized by Reviewer II) and in a similar manner to the approach in a recently published paper (von Steinbuechel et al., 2023; https://doi.org/10.1371/journal.pone.0280796). The literature table is then referenced in the introduction part:

“Table OS1 in the Online Supplement (OS-1 Literary Review) provides an overview of the literature review of factors associated with psychopathological outcomes in adult civilians after TBI [von Steinbuechel et al., 2023].” (lines 77-79).

  1. Sample characteristics move in a methods part with the title – sample...

Response: We understand your request and see some benefit in presenting the Sample Characteristics right after the Participant Data. However, we have followed the Consensus-based Standards for the selection of health Measurement Instruments (COSMIN) reporting guidelines for patient-reported outcome measures (https://www.cosmin.nl/wp-content/uploads/COSMIN-reporting-guideline_1.pdf, last access on 24.05.23) which suggest that Patient Characteristics should be included in the Results Section. We would like to be consistent with this structure. Since the factors presented in the Sample Characteristics are introduced in “2.3.1. Sociodemographic, Premorbid, and Injury-Related Factors” and “2.3.2. PROM Screenings for Psychopathological Symptoms”, we believe that the table should be presented at some point after these subsections. Therefore, we kindly ask for your understanding that we do not integrate your suggestion.

  1. In the results part, I would suggest you move the regression part to the beginning of the paper;

Response: The regression part follows now directly after reporting the descriptive statistics. “Correlations between PROM scores across Psychopathological Domains”, “Correlations between Three-Month and Six-Month PROM Scores”, and “Endorsement of Psychopathological Symptoms” were moved to the Online Supplement (OS-2 Additional Methods Results).

  1. Please, improve the conclusion part with more scientific and social implications;

Response: Thank you very much for this suggestion. We have updated the conclusion subsection with possible scientific and social implications drawn from the results of the current study:

“The scientific implications of the present study include investigating influential factors with regard to several types of assessment (e.g., screening diagnoses, symptoms, PROM scores) of psychopathological outcomes after TBI. In addition, alternative statistical models (e.g., multivariate zero-inflated negative binomial regression) should become technologically and commercially available. On the basis of our results, societal and clinical implications include targeted mental health programs for particularly vulnerable populations after TBI (e.g., females, low educational status), close psychopathological monitoring of TBI patients with a history of psychiatric symptoms, and enhancement of safety features to prevent road traffic accidents with TBI (e.g., collision avoidance systems, airbag technology).“ (lines 561-570).

The scientific implications of the present study include investigating influential factors with regard to several types of assessment (e.g., screening diagnoses, symptoms, PROM scores) of psychopathological outcomes after TBI. In addition, alternative statistical models (e.g., multivariate zero-inflated negative binomial regression) should become technologically and commercially available. On the basis of our results, societal and clinical implications include targeted mental health programs for particularly vulnerable populations after TBI (e.g., females, low educational status), close psychopathological monitoring of TBI patients with a history of psychiatric symptoms, and enhancement of safety features to prevent road traffic accidents with TBI (e.g., collision avoidance systems, airbag technology).

Minor editing of English language required

Response: The manuscript was language edited by a native English speaker.

Thank you very much for your time and helpful suggestions!

Reviewer 2 Report

Hi,

my main suggestion with your paper is to shorten it.  I found it lost its impact by being long and overly detailed.  I suggest you 'eyeball' your methods, results and discussion to find places to improve.  One example, is the methods section which spends a lot of time justifying your statistical approach.  The results are data rich but actually very hard for the average reader to digest.  Another example is your limitations section which is almost a page without breaks.  For the average reader (I am one), I would prefer the paper to be shortened by a third and more selective in reporting.  

Some small points follow

Line 46...If you are reporting incidence, should you specify a time frame eg annually?

Line 48...Aren't all assaults violent?  Is violent needed?

Line 50...which rather than whom.

Line 60...surely you are not advocating for continuous assessment?  Repeated perhaps?

Line 62...I'm not sure that I agree that "as soon as possible" treatment is always required...medications have side effects and poor response rates, therapy is expensive.  The natural history of mental disorder is often to improve after the triggering event...

Figure 1...Need to have a key to the abbreviations

Line 314...40% went to ICU but 60% had mild TBI.   Need to specify that ICU may not be head injury related?

Line 317...I'd avoid words like 'notable' in results and leave for discussion.

Results: as stated early, these are lengthy and the detail means the main findings get lost.

Discussion..."low but notable''...perhaps these need to be bench marked.  Are they higher than general populations?  Is this just expected rates of mental illness in the community?

Some use of language lacked precision.  I noted some of the examples above.  Overall, no concern.

Author Response

Answer to Reviewer II concerning the manuscript

Influence of Sociodemographic, Premorbid, and Injury-Related Factors on Post-traumatic Stress, Anxiety, and Depression after Traumatic Brain Injury

my main suggestion with your paper is to shorten it.  I found it lost its impact by being long and overly detailed. I suggest you 'eyeball' your methods, results and discussion to find places to improve. One example, is the methods section which spends a lot of time justifying your statistical approach. The results are data rich but actually very hard for the average reader to digest. Another example is your limitations section which is almost a page without breaks.  For the average reader (I am one), I would prefer the paper to be shortened by a third and more selective in reporting. 

Response: Dear reviewer, thank you very much for your feedback. Following your suggestion, we have condensed the paper in several sections (e.g., discussion, limitations) and moved parts of the methods and results, as well as selected figures and tables to the Online Supplement (OS-2 Additional Methods Results). To account for the reduced information on the methodological approaches, we have added a table presenting characteristics of the regression models applied in the current study (Table 1). In sum, we were able to reduce the overall length of the manuscript from 819 lines to 573 lines.

Some small points follow

Line 46...If you are reporting incidence, should you specify a time frame eg annually?

Response: We specified the time frame for the reported incidence rate “per year” (line 47).

Line 48...Aren't all assaults violent?  Is violent needed?

Response: Thank you. We removed it (line 49).

Line 50...which rather than whom.

Response: We exchanged “whom” with “with” (line 50).

Line 60...surely you are not advocating for continuous assessment?  Repeated perhaps?

Response: Thank you for your comment. We exchanged the term “continuous” with “repeated” (line 60).

Line 62...I'm not sure that I agree that "as soon as possible" treatment is always required...medications have side effects and poor response rates, therapy is expensive.  The natural history of mental disorder is often to improve after the triggering event...

Response: Thank you very much for giving us the chance to clarify this issue. We wanted to underline the importance of an early detection of psychopathological symptoms and not necessarily a treatment as soon as possible. That way, patients could be more closely monitored and, where indicated, could be offered specific therapeutic interventions after discussing possible side effects or treatment costs. We hope the following reformulated lines are clearer:

“Repeated psychiatric and psychopathological assessments starting in the acute injury stage are therefore essential in order to detect signs of emotional disturbance following TBI as soon as possible. Where indicated, patients could then be offered specific psychological and/or medical/pharmaceutical therapeutic interventions.” (lines 60-64).

Figure 1...Need to have a key to the abbreviations

Response: We have added explanations of the abbreviations in the note to Figure 1 (lines 135-138).

Line 314...40% went to ICU but 60% had mild TBI.   Need to specify that ICU may not be head injury related?

Response: Thank you for pointing out this issue. Indeed, the clinical care pathway, especially admission to the ICU was also related to the treatment of extracranial injuries. To clarify we have added the following pieces of information to the subsections “2.5.1. Sociodemographic, Premorbid, and Injury-Related Factors” and “3.1. Sample Characteristics”:

“Finally, medical records were used to assess information on the injury cause; loss of consciousness (LOC) during the TBI event; and the clinical care pathway where the TBI and/or extracranial injuries were treated (i.e., emergency room, ward, ICU).” (lines 185-187).

The participants were predominantly male (65.00%) and had a mean age of 49.09 years (SD=19.32). While only a minority of participants sustained a moderate/severe TBI (22.81%), individuals were frequently admitted to the ICU (40.60%). This was likely due to the pronounced extracranial injuries (Total ISS, M=18.58, SD=15.00), since individuals with high extracranial injuries (Total ISS15) most commonly received treatment in the ICU (77.81%). Most participants showed good recovery (64.57%).” (lines 237-243).

Line 317...I'd avoid words like 'notable' in results and leave for discussion.

Response: We agree that the use of the term “notable” was not appropriate and exchanged it with “subgroup (line 243) and “moderate” (lines 27-28; line 406; line 523).

Results: as stated early, these are lengthy and the detail means the main findings get lost.

Response: We shortened the results section by moving the subsections “Correlations between PROM scores across Psychopathological Domains”, “Correlations between Three-Month and Six-Month PROM Scores”, and “Endorsement of Psychopathological Symptoms”, as well as parts of ““Regression Models” to the Online Supplement (OS-2 Additional Methods Results). As stated above, we have added a table presenting characteristics of the regression models applied in the current study (Table 1) to account for the reduced information on the methodological approaches (lines 211-229).

Furthermore, now report the range of effect sizes of influential factors instead of all effects individually the subsection “Regression Models”(lines 287-288, 323, 340-341, 366-367). Readers interested in the specific effects of factors are referred to Tables A1-A3, B1-B3, and C1-C3 in the Appendix.

Discussion..."low but notable''...perhaps these need to be bench marked. Are they higher than general populations?  Is this just expected rates of mental illness in the community?

Response: In order to provide a relevant benchmark, we added information on previously reported incidence rates for psychopathological symptoms after TBI to the introduction (lines 72-73) and discussion part (line 408). Since the incidence rate in the current study was rather low compared to previously reported incidence rates, we pointed this out in the discussion part:

“The study sample as a whole showed moderate levels of PTSD, GAD, and MDD at six months after TBI (10.78%—17.74%), which were somewhat lower than previously reported incidence rates (16.5%—24.5%) [Rogers and Read, 2007]” (lines 406-408).

Some use of language lacked precision. I noted some of the examples above.  Overall, no concern

Response: The manuscript was language edited by a native English speaker.

Thank you very much for your time and helpful suggestions!

Round 2

Reviewer 1 Report

Dear authors,

thank you for your contributions.

Kind regards

Author Response

Answer to Reviewer I concerning the manuscript

Influence of Sociodemographic, Premorbid, and Injury-Related Factors on Post-traumatic Stress, Anxiety, and Depression after Traumatic Brain Injury

Dear authors,

thank you for your contributions.

Kind regards

Response: Dear reviewer, once again thank you very much for your time and helpful suggestions!

Fabian Bockhop (first author) and Nicole v. Steinbüchel (last and corresponding author)
